# Helen: Optimizing CTR Prediction Models with Frequency-wise Hessian Eigenvalue Regularization

Submission Id: 880

## ABSTRACT

Click-Through Rate (CTR) prediction holds paramount significance in online advertising and recommendation scenarios. Despite the proliferation of recent CTR prediction models, the improvements in performance have remained limited, as evidenced by open-source benchmark assessments. Current researchers tend to focus on developing new models for various datasets and settings, often neglecting a crucial question: What is the key challenge that truly makes CTR prediction so demanding?

In this paper, we approach the problem of CTR prediction from an optimization perspective. We explore the typical data characteristics and optimization statistics of CTR prediction, revealing a strong positive correlation between the top hessian eigenvalue and feature frequency. This correlation implies that frequently occurring features tend to converge towards sharp local minima, ultimately leading to suboptimal performance. Motivated by the recent advancements in sharpness-aware minimization (SAM), which considers the geometric aspects of the loss landscape during optimization, we present a dedicated optimizer crafted for CTR prediction, named Helen. Helen incorporates frequency-wise Hessian eigenvalue regularization, achieved through adaptive perturbations based on normalized feature frequencies.

Empirical results under the open-source benchmark framework underscore Helen's effectiveness. It successfully constrains the top eigenvalue of the Hessian matrix and demonstrates a clear advantage over widely used optimization algorithms when applied to seven popular models across three public benchmark datasets on BARS. We release our implementation of Helen here[1].

**ACM Reference Format:**
Anonymous Author(s). 2024. Helen: Optimizing CTR Prediction Models with Frequency-wise Hessian Eigenvalue Regularization. In *WWW '24: The Web Conference 2024, May 13–17, 2024, Singapore.* ACM, New York, NY, USA, 12 pages. https://doi.org/10.1145/nnnnnnn.nnnnnnn

## 1 INTRODUCTION

Click-through rate (CTR) prediction holds significant importance in the realm of online advertising and marketing [58, 59, 80]. CTR provides insights into the effectiveness of advertising campaigns by measuring the ratio of users who click on a specific link or

[1]https://anonymous.4open.science/r/Helen-D251

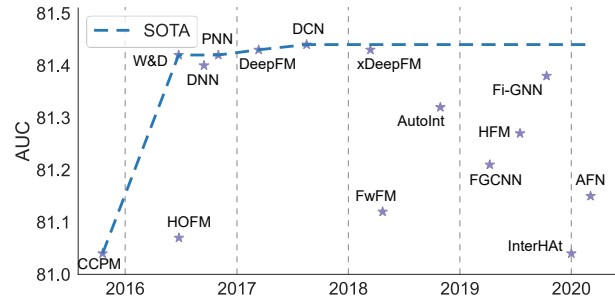

**Figure 1: Evolution of AUC Performance in CTR Prediction Models on the Criteo Dataset, with Time Progression Indicated on the X-axis (Reported by BARS [81]).**

advertisement to the total number of users exposed to it. Accurate CTR prediction is crucial for advertisers as it enables them to assess the performance of their campaigns, allocate budgets effectively, and optimize ad creatives to maximize return on investment (ROI) [42, 58, 80]. For companies with extremely large user bases like Facebook and Google, even a slight improvement in AUC can result in a significant increase in revenue [39, 66, 81].

The pursuit of accurate CTR prediction has garnered substantial attention from various stakeholders since the early 2000s [42, 47, 56]. Efforts to develop sophisticated models to improve CTR prediction accuracy have been persistent in both academic and industrial communities [7, 13, 21, 38, 66, 67, 78, 80]. However, as depicted by Figure 1, despite the surge of new CTR prediction models and architectures, they continue to encounter challenges in improving the performance on benchmark settings.

Besides model structure, the choice of optimization algorithms also has a significant impact on model performance [55, 73]. Adaptive learning rate techniques adpoted by Adam [33] and AdamW [45] have consistently outperformed classic methods such as SGD [57] and its variants [49, 53] over various machine learning tasks and reigned over the field of optimization for the past decade.

Yet, recent attention has been directed towards the customization of optimization algorithms to suit specific tasks and new models, shining a vibrant spotlight on the possibility of inventing new optimizers to further improve model performance. For example, Lion [11] is developed through automated program search and empirically showcased to perform well in diverse computer vision tasks, and Sophia [41] demonstrates distinctive expertise in training large language models (LLM) by employing a lightweight estimate of the diagonal Hessian. Notably, Sharpness-Aware Minimization (SAM) [20] demonstrates that decreasing the sharpness of the loss function will increase models' ability to generalize.

Amidst this rise in recognition of task-tailored optimization methodologies, the field of optimization in CTR prediction has received relatively limited attention. Diverging from many other machine learning tasks, CTR prediction stands out due to its high-dimensional, skewed distribution of categorical features. Crafting a customized optimization approach for this task necessitates special attention to these characteristics.

This paper presents a thorough exploration of how feature frequency influences the optimization process in CTR prediction models. Our investigation reveals a noteworthy correlation: embeddings of more frequently occurring features tend to converge towards sharper local minima, as evidenced by a larger top eigenvalue of the Hessian matrix. In response to these findings, we introduce Helen, a specialized optimizer designed for CTR prediction models. Helen incorporates frequency-wise Hessian eigenvalue regularization, drawing inspiration from SAM, which introduces perturbations to the updating gradient. Our contributions can be summarized as follows:

- We are the first to unveil a robust positive correlation between feature frequency and the top eigenvalue of feature embeddings. This correlation highlights an imbalanced distribution of loss sharpness across the parameter space, making it challenging for the optimizer to discover flat minima that generalize effectively.
- We introduce a specialized optimizer for CTR prediction models, known as Helen. Helen leverages frequency information to estimate the sharpness of feature embeddings and adjusts the perturbation radius accordingly, drawing inspiration from SAM, which has been proven to possess the ability to regularize Hessian eigenvalues.
- With thorough testing under an open-source benchmark setup, we provide empirical evidence across three public datasets and seven established CTR prediction models. The results consistently demonstrate Helen's effectiveness in regularizing the sharpness of the loss function, thereby significantly enhancing the performance of CTR prediction models.

## 2 PRELIMINARIES

### 2.1 CTR Prediction

Let $\mathcal{S} = \{(x_i, y_i)\}_{i=1}^{n}$ represent the training dataset, where each sample $(x_i, y_i)$ follows the distribution $\mathcal{D}$ and $y_i \in \{0, 1\}$ denotes whether the user clicked or not, $x_i = [x_i^1, x_i^2, \ldots, x_i^m]$ encodes categorical information regarding the user, the product, and their interaction, where $m$ indicates the number of feature fields. The $j$-th categorical field is converted through one-hot encoding into a vector denoted as $x_i^j \in \{0, 1\}^{s_j}$, where $s_j$ represents the number of total feature within the $j$-th categorical field and $x_i^j[k] = 1$ only if the $k$-th feature in field $j$ is present in the $i$-th sample.

Given the predictive network denoted as $f$, predictions are generated using the function $f(x; w)$, where $w = [h, e]$ comprises two parts: $e$ for the embeddings of sparse features and $h$ for all remaining dense network's hidden layers. The embedding weights $e$ can be further subdivided into $m$ parts, represented as $e = [e^1, e^2, \ldots, e^m]$, where $e^j$ signifies the embedding weights associated with the $j$-th field. Within each field, the embedding weights $e^j$ can be broken down into $s_j$ components, denoted as $e^j = [e_1^j, e_2^j, \ldots, e_{s_j}^j]$, with

$e_k^j$ representing the weights for the $k$-th feature in the $j$-th field. Assume that $h \in \mathbb{R}^{d_h}$ and $e_k^j \in \mathbb{R}^{d_e}$, where $d_h$ and $d_e$ denote the dimensions of the dense network and embeddings, respectively.

### 2.2 Model Optimization

Given the predicting network $f$, for a specific sample $(x, y)$ and model parameter $w$, the loss function $\mathcal{L}(x, y, f(x; w))$ measures the difference between the prediction $f(x; w)$ and the ground truth $y$. The loss function $\mathcal{L}$ is usually defined as the cross-entropy loss for CTR prediction task.

The ultimate goal of model optimization is to solve the following optimization problem:

$$w^* = \arg\min_{w} \mathbb{E}_{(x,y)\sim\mathcal{D}} \mathcal{L}(x, y, f(x; w)), \tag{1}$$

where $w^*$ is the optimal model parameter. However, this optimization problem is intractable since the distribution $\mathcal{D}$ is unknown. Instead, we can solve the following empirical risk minimization problem based on the training dataset $\mathcal{S}$:

$$\hat{w} = \arg\min_{w} \frac{1}{n} \sum_{i=1}^{n} \mathcal{L}(x_i, y_i, f(x_i; w)). \tag{2}$$

### 2.3 Sharpness-Aware Minimization

The objective of the SAM algorithm [20] is to identify the parameters that minimize the training loss $\mathcal{L}_{\mathcal{S}}(w)$ while considering neighboring points within the $\ell_p$ ball. This is achieved through the utilization of the following modified objective function:

$$\mathcal{L}_{\mathcal{S}}^{SAM}(w) = \max_{\|\epsilon\|_p \leq \rho} \mathcal{L}_{\mathcal{S}}(w + \epsilon), \tag{3}$$

where $|\epsilon|_p \leq \rho$ ensures that the magnitude of the perturbation $\epsilon$ remains within a specified threshold.

As calculating the optimal solution of inner maximization is infeasible, SAM employs a one-step gradient to approximate the maximization problem, as demonstrated below:

$$\hat{\epsilon}(w) = \rho \frac{\nabla_w \mathcal{L}_{\mathcal{S}}(w)}{\|\nabla_w \mathcal{L}_{\mathcal{S}}(w)\|} \approx \arg\max_{\|\epsilon\|_p \leq \rho} \mathcal{L}_{\mathcal{S}}(w + \epsilon). \tag{4}$$

Finally, SAM computes the gradient with respect to perturbed model $w + \hat{\epsilon}$ for the update:

$$g = \nabla_w \mathcal{L}_{\mathcal{S}}^{SAM}(w) \approx \nabla_w \mathcal{L}_{\mathcal{S}}(w)|_{w+\hat{\epsilon}}. \tag{5}$$

## 3 METHOD

We begin by establishing the key characteristics of the CTR prediction task in Section 3.1, primarily focusing on skewed feature distributions. Subsequently, we analyze the intricate connection between feature frequencies and the Hessian matrix in Section 3.2.

Drawing from our analytical insights, we introduce Helen, a novel and tailored optimizer designed specifically for CTR prediction models in Section 3.3.

### 3.1 Skewed Feature Distribution

A prominent feature that sets CTR prediction apart from other traditional machine learning tasks, such as image classification, is its distinctive input data format—often comprised of multi-hot encoded categorical features. Within CTR prediction models, the possible

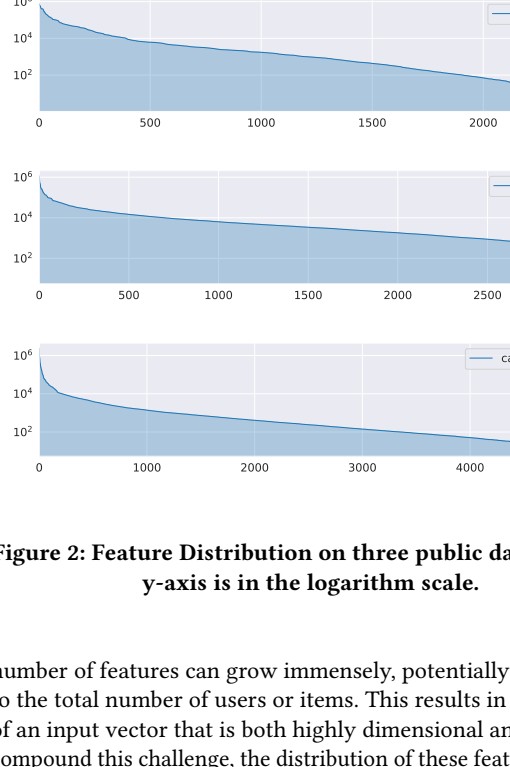

**Figure 2: Feature Distribution on three public datasets. The y-axis is in the logarithm scale.**

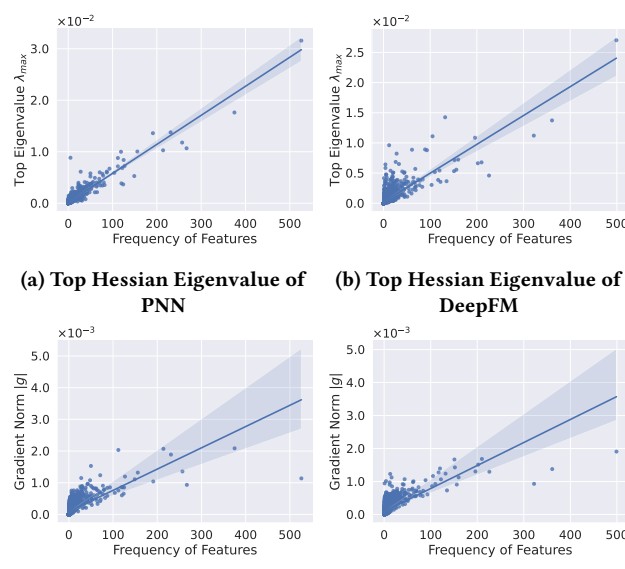

(a) Top Hessian Eigenvalue of PNN

(b) Top Hessian Eigenvalue of DeepFM

(c) Gradient Norm of PNN

(d) Gradient Norm of DeepFM

**Figure 3: PNN and DeepFM trained on Criteo dataset with Adam optimizer.**

number of features can grow immensely, potentially comparable to the total number of users or items. This results in the creation of an input vector that is both highly dimensional and sparse. To compound this challenge, the distribution of these features exhibits a significant skew. As shown in Figure 2, the distribution of feature frequency is highly skewed. Popular features are much more frequent than unpopular features.

Many previous works [3, 50, 72, 74] have realized the challenge caused by the skewed feature distribution and propose different techniques such as counterfactual reasoning [5, 40, 74, 76] and model regularization [2, 30, 61]. However, these works slide over the intrinsic difficulty that a skewed feature distribution brings to the optimization of CTR prediction models, which leads to sub-optimal recommendation performance.

A recent work, CowClip [77], proposes to clip the gradient of feature embedding according to the frequencies of features and successfully scales the batch size to reduce the training time. Cowclip first considers the impact of skewed feature distribution from the perspective of optimization. However, its discussion is limited to the first-order gradient and fails to unveil the second-order information of the loss landscape, *i.e.,* Hessian matrix.

In the following section, we will show that even if feature frequencies do affect the feature embedding gradient norm, the impact is not as significant as the correlation between feature frequency and the dominant eigenvalue of the Hessian matrix.

## 3.2 Hessian and Frequency of Features

Previous works [36, 51, 60] have shown that under sufficient regularity conditions, gradient-based methods are guaranteed to converge to local minimizer $w^*$ of the loss function, such that

$$g = \nabla_w \mathcal{L}_S(w^*) = 0.$$

and the Hessian matrix $H = \nabla_w^2 \mathcal{L}_S(w^*)$ is positive semi-definite, *i.e.,* all eigenvalues of $H$ are non-negative.

Bottou [6] demonstrates that the largest eigenvalue $\lambda$ of the Hessian matrix $H$ of the loss function $\mathcal{L}_S(\cdot)$ indicates whether the optimization is ill-conditioned. If the dominant eigenvalue is too large, the gradient-based optimization methods will converge to some deep ravines in the loss landscape, which leads to poor generalization performance [8, 26, 31, 32].

In the context of CTR prediction, we delve deeper into examining the connection between feature frequencies and the dominant eigenvalues of the Hessian matrix associated with the respective feature embedding. The frequency of each feature $k$ in the $j$-th feature field is counted according to the following equation,

$$N_k^j(S) = \sum_{i=1}^{n} x_i^j[k]. \tag{6}$$

As discussed in Section 2.1, the parameters of CTR prediction models can typically be categorized as

$$w = [h, \underbrace{e_1^1, e_2^1, \ldots, e_{s_1}^1}_{\text{1-st Field}}, \underbrace{e_1^2, e_2^2, \ldots, e_{s_2}^2}_{\text{2-nd Field}}, \ldots, \underbrace{e_1^m, e_2^m, \ldots, e_{s_m}^m}_{m\text{-th Field}}].$$

Similarly, the gradient can be decomposed as:

$$g = [g_h, \underbrace{g_1^1, g_2^1, \ldots, g_{s_1}^1}_{\text{1-st Field}}, \underbrace{g_1^2, g_2^2, \ldots, g_{s_2}^2}_{\text{2-nd Field}}, \ldots, \underbrace{g_1^m, g_2^m, \ldots, g_{s_m}^m}_{m\text{-th Field}}].$$

Regarding the Hessian matrix $H$, our primary focus centers on the diagonal block matrix $diag(H)$, which is defined as:

$$[H_h, \underbrace{H_1^1, H_2^1, \ldots, H_{s_1}^1}_{\text{1-st Field}}, \underbrace{H_1^2, H_2^2, \ldots, H_{s_2}^2}_{\text{2-nd Field}}, \ldots, \underbrace{H_1^m, H_2^m, \ldots, H_{s_m}^m}_{m\text{-th Field}}].$$

Here, the diagonal block matrix $H_h$ is a $d_h \times d_h$ matrix and $H_k^j$ is a $d_e \times d_e$ matrix for $j \in \{1, 2, \ldots, m\}$ and $k \in \{1, 2, \ldots, s_j\}$. Given

**Table 1: Correlation between the norm of the Hessian matrix and the frequency of features.**

| Dataset | PNN | | DeepFM | |
|---------|-----|-----|--------|-----|
| | $r(\|g\|, N)$ | $r(\lambda, N)$ | $r(\|g\|, N)$ | $r(\lambda, N)$ |
| Avazu | 0.758 | 0.841 | 0.759 | 0.809 |
| Criteo | 0.715 | 0.943 | 0.700 | 0.825 |
| Taobao | 0.806 | 0.990 | 0.824 | 0.977 |

our primary focus on the feature distribution, we use the notation $\lambda_k^j$ to represent the top eigenvalue of the Hessian matrix $H_k^j$.

We choose two popular CTR prediction models, PNN [54] and DeepFM [21], and train them with Adam [33] optimizer. Figures 3a and 3b visually represent the relationship between the highest eigenvalue of the Hessian matrix, $\lambda_k^j$, and the frequency of the corresponding feature $N_k^j$ within the feature field labeled as 'C13' in the Criteo dataset. To offer a point of comparison, we also include graphical representations of the gradient norm of feature embedding $\left|g_k^j\right|$ plotted against the frequency of the corresponding feature $N_k^j$ in Figures 3c and 3d.

When comparing the correlation between the gradient norm of feature embeddings and feature frequency to the correlation involving the top eigenvalue of the diagonal block of the Hessian matrix, a striking difference emerges. The latter exhibits a considerably more pronounced association, signifying an exceptionally strong and positive linear relationship.

To further substantiate this observation, we conduct an analysis by computing the Pearson correlation coefficient $r(\lambda, N)$ between the top eigenvalue of the diagonal block in the Hessian matrix and the corresponding feature's frequency. The results, as depicted in Table 1, also include the Pearson correlation coefficient for the gradient norm and feature frequency.

Across the three benchmark public datasets, it is evident that the correlation coefficient, denoted as $r(\lambda, N)$, between the top eigenvalue of the diagonal Hessian block and the corresponding feature's frequency consistently exceeds 0.8. This value signifies a nearly perfect positive linear relationship between these two variables.

This strongly implies that features with higher frequencies are more likely to converge to sharper local minima.

## 3.3 Helen: Frequency-wise Hessian Eigenvalue Regularization

SAM [20] has proven its efficacy in enhancing the generalization performance of deep learning models by concurrently minimizing both the loss value and sharpness. Recent advancements in both experimental studies [10, 20] and theoretical research [68] have established SAM's ability to effectively reduce the dominant eigenvalue of the Hessian matrix.

LEMMA 1. *Minimizing the SAM loss function*

$$\mathcal{L}_S^{SAM}(w) = \max_{\|\epsilon\|_p \leq \rho} \mathcal{L}_S(w + \epsilon)$$

*introduces a bias*

$$\arg\min_w \lambda\left(\nabla_w^2 \mathcal{L}_S(w)\right)$$

*among the minimizers of the original loss $\mathcal{L}_S(w)$ in an $O(\rho)$ neighborhood manifold, where $\lambda\left(\nabla_w^2 \mathcal{L}_S(w)\right)$ denotes the maximum eigenvalue of the Hessian matrix.*

In particular, Wen et al. [68] demonstrates that SAM diminishes the largest eigenvalue of the Hessian matrix of the loss function in the local vicinity of the manifold bounded by the perturbation radius $\rho$, as summarized in Lemma 1. This finding has prompted our exploration into the regularization of the Hessian matrix for feature embedding using SAM.

However, a limitation of the native SAM lies in its application of a uniform perturbation radius $\epsilon$ to all model parameters. As previously demonstrated in Section 3.1 and Section 3.2, we have underscored the notable skew in the distribution of feature frequencies and the robust correlation existing between these frequencies and the top eigenvalue of the Hessian matrix.

When the uniform perturbation radius $\epsilon$ is relatively small, it inadequately regularizes Hessian eigenvalues for high-frequency features, leading to suboptimal performance, which deviates from our goals. Conversely, when utilizing a large uniform perturbation radius $\epsilon$, the top eigenvalue of the Hessian matrix for features with higher frequencies decreases, indicating a convergence toward flatter local minima. However, this strategy overly regularizes features with lower frequencies, redirecting their optimization toward flat regions at the cost of refining the original loss function $\mathcal{L}_S(w)$. This trade-off between high-frequency and low-frequency features ultimately leads to a suboptimal solution.

Inspired by the aforementioned insights, we propose Helen, a novel optimization algorithm with frequency-wise Hessian eigenvalue regularization. Helen is founded on the SAM with an adaptive perturbation radius $\epsilon$ for each feature embedding.

During the training process, Helen first calculates the frequency of each feature $k$ in any feature field $j$, denoted as $N_k^j$. Then the perturbation radius $\rho_k^j$ is calculated as follows,

$$\rho_k^j = \rho \cdot \max\{\frac{g_k^j}{\|g_k^j\|}, \xi\}. \tag{7}$$

Given the relatively small proportion of infrequent features in relation to the most frequent feature, we have introduced a lower-bound parameter denoted as $\xi$ to avoid setting the perturbation radius $\rho_k^j$ to excessively small values.

And then approximate Equation 3 with first-order Taylor expansion, we have

$$\hat{\epsilon}(e_k^j) = \arg\max_{\|\epsilon\|_p \leq \rho*_k} \mathcal{L}_S(e_k^j + \epsilon) \approx \rho_k^j \cdot \frac{\nabla_{e_k^j} \mathcal{L}_S(w)}{\|\nabla_{e_k^j} \mathcal{L}_S(w)\|}. \tag{8}$$

For the dense network parameters $h$, we use the same perturbation radius $\rho$ as SAM.

After we have all the perturbation vectors

$$\hat{\epsilon}(w) = [\hat{\epsilon}(h), \hat{\epsilon}(e_1^1), \ldots, \hat{\epsilon}(e_{s_m}^m)], \tag{9}$$

we can calculate the Helen gradient with respect to perturbed model $w + \hat{\epsilon}$ similar with Equation 5 for the update:

$$g^{Helen} = \nabla_w \mathcal{L}_S(w)|_{w + \hat{\epsilon}(w)}. \tag{10}$$

The culmination of our work yields the ultimate Helen algorithm, achieved by implementing a conventional numerical optimizer, such as stochastic gradient descent (SGD), and replacing the conventional gradient with the Helen gradient. Algorithm 1 provides pseudocode for the complete Helen algorithm, with SGD as the underlying optimization method.

## 4 EXPERIMENTS

### 4.1 Experimental Setup

*4.1.1 Datasets.* We evaluate the proposed optimizer on three benchmark public datasets, namely Avazu, Criteo, and Taobao. For more statistics of these three datasets , please refer to Appendix A.2. A brief introduction to them is as follows:

- **Avazu**: The Avazu dataset [4] employed in this research comprises approximately 10 days of labeled click-through data associated with mobile advertisements.
- **Criteo**: The Criteo dataset [35] serves as a prominent benchmark dataset widely utilized for CTR prediction, specifically in the context of display advertising.
- **Taobao**: The Taobao dataset [64] comprises a collection of advertisement display and click records, which were randomly selected from Taobao over a span of 8 days.

For a comprehensive and equitable comparison, we use the datasets preprocessed by the BARS benchmark [81]. In particular, we adopt the officially recommended 'Criteo_x4_001' for the Criteo dataset and the 'Avazu_x4_001' for the Avazu dataset among all the available preprocessed versions.

*4.1.2 CTR Prediction Models.* To assess Helen's performance, we selected seven well-established CTR prediction models, specifically DNN [13], WideDeep [12], PNN [54], DeepFM [21], DCN [66], DLRM [48], and DCNv2 [67]. The choice of these models is primarily influenced by two key considerations: their impact within both the academic and industry communities and their high performance on the BARS benchmark leaderboard. Each of these seven models has accumulated over 200 citations and has consistently showcased robust performance on the BARS benchmark. For further details of these models, please refer to Appendix A.5.

*4.1.3 Baselines.* To demonstrate the effectiveness of our proposed method, we compare Helen with seven commonly used optimizers, namely Adam [33], Nadam [16], Radam [43], SAM [20], and ASAM [34]. Nadam and Radam are both variants of Adam, and ASAM is an adaptive version of SAM. For more details, please refer to Appendix A.6

*4.1.4 Implementation details.* Our experiment is conducted using FuxiCTR[2] [81, 82], an open-source framework for CTR prediction. We strictly adhere to the benchmark's coding standards for CTR model implementation, data processing, and model evaluation. In our experiment, we rely on the official PyTorch implementations of

Adam, Nadam, Radam, and an open-source PyTorch implementation[3] of SAM. For ASAM, we employ the official implementation[4] provided by the authors.

Regarding model hyperparameters, we adopt the benchmark settings reported by FuxiCTR and keep them constant for all the optimizers. As for optimizer hyperparameters, the learning rate remains fixed at $\eta = 1 \times 10^{-3}$ since all the tested optimizers incorporate an adaptive learning rate mechanism and are not particularly sensitive to changes in this parameter. In terms of L2-regularization coefficients, we explore values from the set $\{1 \times 10^4, 1 \times 10^{-5}, 0\}$, maintaining consistency with the search space employed in BARS. For SAM, ASAM, and Helen, we perform tuning on the perturbation radius $\rho$, considering values from the set $\{0.05, 0.01, 0.005, 0.001\}$. In the case of Helen, the lower bound $\xi$ is varied within $\{0, 0.5\}$.

### 4.2 Performance Analysis

We evaluate the Helen optimizer alongside Adam, Nadam, Radam, SAM, and ASAM using seven CTR prediction models across three datasets. The summarized outcomes for the Avazu and Taobao datasets are presented in Table 2 and Table 3, respectively. For results of Taobao, please refer to Appendix A.3. The analysis of these tables leads to the following insights:

*4.2.1 **Narrow Performance Disparity Among CTR Prediction Models***. While a multitude of CTR prediction models have emerged since the advent of DNN in 2016, proclaiming their superiority over predecessors, the actual performance disparity among these models remains relatively modest. The performance gap between the best and worst-performing models in terms of AUC stands at 2.30%, 0.13%, and 0.61% for the Avazu, Criteo, and Taobao datasets, respectively.

When excluding the highest and lowest AUC scores, this performance gap further narrows to 0.35%, 0.07%, and 0.51% in the Avazu, Criteo, and Taobao datasets, respectively. Notably, model performance exhibits inconsistency across different datasets. For instance, DCNv2 excels in the Criteo dataset but ranks as the least effective model in the Avazu dataset, showcasing a significant performance contrast relative to other models.

Among the seven models evaluated, PNN and DeepFM demonstrate the highest stability, with PNN claiming the top position in two datasets and DeepFM consistently outperforming the average in all three datasets.

*4.2.2 **Limited Performance Enhancement with Adam Variants***. Although Nadam and Radam are theoretically proven to have superior convergence bounds or lower variance, their actual performance improvements are rather limited. The average enhancement over seven models achieved by Nadam compared to Adam amounts to a mere 0.15%, 0.03%, and -0.09% in terms of AUC for the Avazu, Criteo, and Taobao datasets, respectively.

Similarly, the average improvement with Radam over Adam stands at 0.03%, 0.03%, and -0.27% in AUC for the Avazu, Criteo, and Taobao datasets, respectively. Notably, the performance of Nadam and Radam exhibits inconsistency across different datasets, proving

---

[2]https://github.com/xue-pai/FuxiCTR

[3]https://github.com/davda54/sam
[4]https://github.com/SamsungLabs/ASAM

**Table 2: Overall performance on the Avazu dataset.**

| Model | Adam | | Nadam | | Radam | | SAM | | ASAM | | Helen | |
|---|---|---|---|---|---|---|---|---|---|---|---|---|
| | LogLoss | AUC | LogLoss | AUC | LogLoss | AUC | LogLoss | AUC | LogLoss | AUC | LogLoss | AUC |
| DNN | 37.302 | 79.271 | 37.365 | 79.142 | 37.399 | 79.027 | **37.269** | 79.265 | 37.294 | 79.275 | 37.271 | **79.279** |
| WideDeep | 37.299 | 79.122 | 37.607 | 78.908 | 37.510 | 78.794 | 37.306 | 79.108 | 37.291 | 79.142 | **37.284** | **79.147** |
| PNN | **37.209** | 79.413 | 37.215 | 79.355 | 37.432 | 79.065 | 37.227 | 79.355 | 37.223 | 79.401 | 37.209 | 79.409 |
| DeepFM | 37.245 | 79.279 | 38.088 | 78.897 | 37.407 | 79.034 | 37.269 | 79.260 | 37.251 | 79.287 | **37.237** | **79.303** |
| DCN | 37.237 | 79.245 | 37.352 | 79.056 | 37.390 | 78.976 | 37.271 | 79.224 | 37.238 | **79.250** | **37.228** | 79.250 |
| DLRM | 37.507 | 78.924 | 37.972 | 79.057 | 37.510 | 78.881 | 37.516 | 78.921 | 37.510 | 78.913 | **37.174** | **79.400** |
| DCNv2 | 38.480 | 77.108 | 37.436 | 79.018 | 37.509 | 78.812 | 38.582 | 76.908 | 38.475 | 77.116 | **37.319** | **79.100** |
| *avg* | 37.468 | 78.909 | 37.576 | 79.062 | 37.451 | 78.941 | 37.491 | 78.863 | 37.469 | 78.912 | **37.246** | **79.270** |

**Table 3: Overall performance on the Criteo dataset.**

| Model | Adam | | Nadam | | Radam | | SAM | | ASAM | | Helen | |
|---|---|---|---|---|---|---|---|---|---|---|---|---|
| | LogLoss | AUC | LogLoss | AUC | LogLoss | AUC | LogLoss | AUC | LogLoss | AUC | LogLoss | AUC |
| DNN | 43.830 | 81.364 | 43.830 | 81.379 | 43.815 | 81.394 | 43.784 | 81.417 | 43.807 | 81.384 | **43.768** | **81.434** |
| WideDeep | 43.820 | 81.375 | 43.804 | 81.388 | 43.783 | 81.410 | 43.771 | 81.418 | 43.808 | 81.394 | **43.784** | **81.421** |
| PNN | 43.888 | 81.332 | 43.805 | 81.407 | 43.804 | 81.408 | 43.809 | 81.392 | 43.873 | 81.348 | **43.780** | **81.402** |
| DeepFM | 43.835 | 81.366 | 43.792 | 81.415 | 43.794 | 81.411 | 43.788 | 81.404 | 43.802 | 81.399 | **43.735** | **81.471** |
| DCN | 43.800 | 81.401 | 43.787 | 81.428 | 43.789 | 81.416 | 43.809 | 81.407 | 43.803 | 81.402 | **43.795** | 81.422 |
| DLRM | 43.931 | 81.277 | 43.924 | 81.288 | 43.928 | 81.269 | 43.936 | 81.278 | 43.974 | 81.242 | **43.812** | **81.382** |
| DCNv2 | 43.813 | 81.411 | 43.807 | 81.418 | 43.786 | 81.436 | 43.788 | 81.438 | 43.803 | 81.422 | **43.734** | **81.468** |
| *avg* | 43.845 | 81.361 | 43.822 | 81.389 | 43.814 | 81.392 | 43.812 | 81.393 | 43.838 | 81.370 | **43.773** | **81.428** |

beneficial in the Avazu and Criteo datasets but detrimental in the Taobao dataset.

*4.2.3 **Performance Impact of SAM in Comparison to Adam**.* Deploying SAM directly to CTR prediction models does not guarantee performance improvement. SAM does yield positive results in the Criteo dataset, with a performance gain of 0.03% in terms of AUC. However, on average, models trained with SAM exhibit slightly lower AUC scores compared to those trained with Adam, resulting in performance declines of 0.04% and 0.10% in the Avazu and Taobao datasets, respectively.

Conversely, ASAM consistently outperforms Adam in all three datasets, with an average performance gain of 0.03‰, 0.01%, and 0.04% in the Avazu, Criteo, and Taobao datasets, respectively. These results underscore the significance of employing an adaptive perturbation radius, as demonstrated by ASAM, to achieve enhanced performance, at least in the context of CTR prediction tasks.

*4.2.4 **Helen's Consistent Superiority Across Three Datasets**.* Helen consistently outperforms Adam, Nadam, Radam, SAM, and ASAM across all three datasets, securing the top position in 20 out of 21 experiments. It delivers an average performance improvement of 0.36%, 0.07%, and 0.37% in terms of AUC for the Avazu, Criteo, and Taobao datasets, respectively. In terms of LogLoss, Helen records an average performance reduction of 0.22%, 0.07%, and 0.37% compared to Adam in the Avazu, Criteo, and Taobao datasets.

Furthermore, we conducted paired $t$-tests to compare the differences in AUC scores across all seven models and three datasets. The null hypothesis ($H_0$) posited no significant difference between the two optimizers. The resulting $p$-value is $8 \times 10^{-3}$, which falls below

the conventional significance threshold of 0.05. Thus, we reject the null hypothesis and conclude that Helen significantly outperforms Adam.

*4.2.5 **Helen's Role in Narrowing the Performance Gap Among CTR Prediction Models**.* As discussed in Section 4.2.1, the performance gap between the best and worst-performing models remains relatively small. A notable instance is evident in the results presented in Table 2, where the convergence of DCNv2 and DLRM to exceedingly sharp local optima creates a significant performance disparity when compared to other models. As shown in Figure 5, Helen effectively addresses this issue, substantially reducing this performance gap, making them comparable to other models.

To elaborate, we calculated the variance of AUC scores across all seven models. The variances for Adam are $6.54 \times 10^{-1}$, $2.03 \times 10^{-3}$, and $6.40 \times 10^{-2}$ for the Avazu, Criteo, and Taobao datasets, respectively. In contrast, Helen's variances are $1.36 \times 10^{-2}$, $1.07 \times 10^{-3}$, and $5.55 \times 10^{-3}$ for the Avazu, Criteo, and Taobao datasets, respectively.

These results indicate that Helen can effectively reduce the performance variance of different models, and uniformly improve the performance of CTR prediction models.

## 4.3 Hessian Eigenspectrum Analysis

In alignment with the motivations delineated in Section 3.2, we conduct a comprehensive exploration to investigate how the relationship between Hessian eigenvalues and feature frequencies evolves when training CTR prediction models with different optimizers, namely Adam, SAM, and Helen.

**Table 4: Comparing the Performance of Helen Variants. Notably, both Helen and SAM introduce perturbations to the embedding weights but Helen is characterized with frequency-wise perturbation radius.**

| Model | Optimizer | Embedding Perturbation | Network Perturbation | Lower Bound | LogLoss | AUC |
|---|---|---|---|---|---|---|
| DeepFM | Adam | ✗ | ✗ | ✗ | 43.835 | 81.366 |
| DeepFM | SAM | ✔ | ✔ | ✗ | 43.809 | 81.407 |
| DeepFM | Helen-$m$ | ✔ | ✗ | ✗ | 43.803 | 81.412 |
| DeepFM | Helen-$b$ | ✔ | ✔ | ✗ | 43.746 | 81.461 |
| DeepFM | Helen | ✔ | ✔ | ✔ | **43.735** | **81.471** |

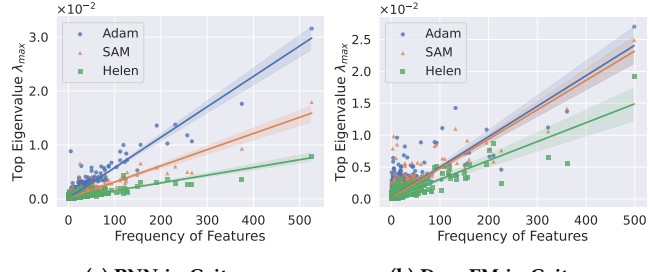

(a) PNN in Criteo      (b) DeepFM in Criteo

**Figure 4: Dominant Hessian eigenvalues of PNN and DeepFM trained with Adam and Helen on the Criteo dataset.**

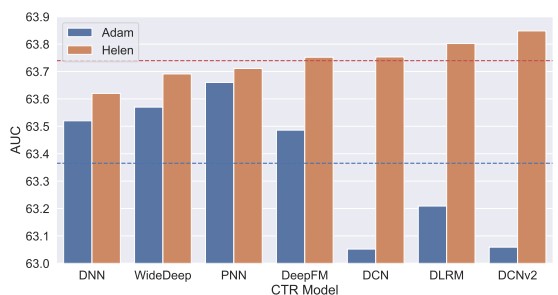

**Figure 5: Helen exhibits a lower variance across various models in comparison to Adam on the Taobao dataset.**

we trained PNN and DeepFM on the Criteo dataset, utilizing each of the three mentioned optimizers—Adam, SAM, and Helen. For the sake of fair comparison, we maintained a uniform perturbation radius ($\rho$) of 0.05 for both SAM and Helen. It's worth emphasizing that, as specified in Equation 7, the perturbation radius $\rho_k^j$ for a specific embedding in Helen is normalized by the maximum frequency, ensures that it does not exceed the value of $\rho$.

*4.3.1 SAM's Effective Regularization of Hessian Eigenvalues.* As depicted in Figure 4, it is evident that SAM effectively reduces the top Hessian eigenvalues across all embeddings. The average top eigenvalue experiences a notable decline, decreasing from $4.45 \times 10^{-4}$ to $2.86 \times 10^{-4}$ in PNN and from $4.59 \times 10^{-4}$ to $4.14 \times 10^{-4}$ in DeepFM. This observation aligns with findings reported in previous studies [10, 20, 68]. Furthermore, a closer examination of

the results in Table 3 affirms that the reduction in the top eigenvalue of the Hessian matrix correlates with an enhancement in the generalization performance of CTR prediction models.

*4.3.2 Helen's Superior Hessian Eigenvalue Regularization.* Remarkably, the results depicted in Figure 4 demonstrate that Helen regularizes the top Hessian eigenvalues more effectively than SAM, with the average top eigenvalue decreasing from $4.45 \times 10^{-4}$ to $1.66 \times 10^{-4}$ in PNN, and from $4.59 \times 10^{-4}$ to $2.67 \times 10^{-4}$ in DeepFM.

Also the standard variance of the top eigenvalues decreases from $1.11 \times 10^{-3}$ to $3.07 \times 10^{-4}$ in PNN, and from $1.06 \times 10^{-3}$ to $6.21 \times 10^{-4}$ in DeepFM. This indicates a more uniform distribution of sharpness across different embeddings. For more details, please refer to the Appendix A.4.

This phenomenon comes from Helen's distinctive perturbation strategy, which places a special emphasis on the frequent features. These frequently occurring features naturally tend to guide the optimization process into sharper local optima, making them substantial contributors to the overall model sharpness. The prioritization of regularization for these frequent features effectively guides the model towards minima that are more generalizable.

We contend that this frequency-wise Hessian eigenvalue regularization plays a pivotal role in driving Helen's superior performance, in perfect alignment with our overarching objective of enhancing generalization.

## 4.4 Ablation Study

In this section, we conduct a comprehensive dissection and evaluation of the individual components and variants comprising the Helen optimizer.

Specifically, we assess the impact of perturbations applied to embedding parameters and network parameters individually as the perturbations to embedding parameters and network parameters can be operated independently. When uniform perturbations are applied to all embedding and network parameters, the Helen optimizer essentially reverts to the SAM optimizer. Additionally, we investigate the role of the lower-bound parameter $\xi$ on the perturbation radius $\rho_k^j$.

We denote the variant of the original Helen optimizer as Helen-$m$ (indicating Helen-minimal), which exclusively implements frequency-wise Hessian eigenvalue regularization for the embedding parameters.

Similarly, we introduce Helen-$b$ (representing Helen-base) as another deviation from the original Helen optimizer. Helen-$b$ combine frequency-wise embedding perturbation in conjunction with

normal SAM perturbation for network parameters, omitting the lower-bound $\xi$ constraint on the perturbation radius $\rho_k^j$.

To maintain simplicity and clarity, our ablation study is exclusively performed on the Criteo dataset, and the results are thoughtfully summarized in Table 4. And we have the following observations:

*4.4.1* **Enhanced Performance Through Feature-wise Hessian Eigenvalue Regularization**. SAM and Helen-*b* both apply perturbations to both embedding and network parameters. Nevertheless, the key distinction between them lies in their perturbation strategies. Helen-*b* employs frequency-wise embedding perturbation, while SAM employs uniform embedding perturbation. The results, as presented in Table 4, unambiguously demonstrate Helen-*b*'s superiority over SAM in terms of both LogLoss and AUC. This compelling evidence underscores the pivotal role of frequency-wise Hessian eigenvalue regularization in enhancing generalization performance.

*4.4.2* **The Significance of Embedding Parameter Perturbation**. While Helen-*m*, exclusively perturbing the embedding parameters, demonstrates its superiority over SAM in both LogLoss and AUC, it is crucial to recognize the added value that regularization of network parameters brings to the table. As evidenced in Table 4, Helen-*b* surpasses Helen-*m* by a substantial margin of 0.5‰ in terms of AUC, showing a notable enhancement in performance. This highlights the importance of perturbing both embedding and network parameters in optimizing model generalization.

## 5 RELATED WORKS

### 5.1 Click-Through Rate Prediction

In the realm of online advertising, metrics like clicks and click-through rate (CTR) serve as indicators of the relevance of advertisements from the users' perspective. It is widely acknowledged by both researchers and industry professionals that improving CTR is essential for the sustainable growth of online advertising ecosystems [58, 59]. As a result, there has been a significant research focus on advertising CTR prediction in recent decades [47, 56, 70].

Early works in CTR and user preference prediction were predominantly based on linear regression (LR) [47] and matrix factorization (MF) [46]. In recent years, deep learning has demonstrated remarkable success in CTR prediction, leading to numerous applications in the field [42, 59, 79].

A typical industrial CTR prediction model, as exemplified in studies such as [69, 75, 79], comprises two main types of parameters: sparse embedding and dense network. However, sparse embedding parameters often account for more than 99% of the total parameter count [77].

### 5.2 Optimizer for Machine Learning

Optimizers for machine learning algorithms could largely influence the performance and convergence of neural networks. SGD [57] is the earliest and also the most representative optimizer for neural network training, which update model parameters based on gradients computed from randomly sampled subsets of training data. More recently, adaptive gradient algorithms have been proposed and widely used in various tasks, such as computer vision [23]

and natural language processing [14]. For example, AdaGrad [18] and RMSProp [24] can dynamically adjust learning rate of each parameter based on their historical gradients. Then, Kingma and Ba proposed Adam, which combines the benefits of both momentum-based and adaptive learning rate approaches for effective optimization. After that, many variants of Adam [16, 43] are proposed and can further improve the performance.

While these methods can achieve great performance in most areas. However, to the best of our knowledge, there is still no optimizer specifically for the CTR task. That also motivates us to design a novel optimizer Helen for the CTR task.

### 5.3 Sharpness and Generalization

The presence of sharp local minima in deep networks can significantly impact their generalization performance [9, 19, 27, 29, 34, 63]. As a result, numerous recent studies have sought to investigate these sharp local minima and address the associated optimization challenges [9, 15, 20, 22, 34, 37, 65, 71].

Recently, Sharpness-Aware Minimization (SAM) [20] presented a novel approach that simultaneously minimizes loss value and loss sharpness to narrow the generalization gap. SAM and its variants demonstrated state-of-the-art performance and achieved rigorous empirical results across various benchmark experiments [1, 17, 34, 44, 83]. However, it is still not clear whether SAM can benefit to CTR task. Therefore, the primary focus of this paper is to further explore the potential of SAM in enhancing the performance of CTR task.

## 6 CONCLUSION

In this study, we have unveiled an intriguing and consistent pattern in CTR prediction models, demonstrating a strong positive correlation between feature frequencies and the top Hessian eigenvalues associated with each feature. Given that the top Hessian eigenvalue serves as a crucial indicator of local minima sharpness, this observation implies that frequent features play a guiding role in steering the optimization process towards sharper local minima.

To leverage this insight, we introduce a novel optimizer, Helen, which prioritizes the regularization of the top Hessian eigenvalues of embedding parameters based on their respective feature frequencies. Through an extensive series of experiments, we have illustrated Helen's remarkable effectiveness, consistently outperforming benchmark optimizers including Adam, Nadam, Radam, SAM, and ASAM across three prominent datasets.

Our in-depth analysis of Hessian eigenvalues has revealed that Helen's frequency-wise Hessian eigenvalue regularization effectively reduces the sharpness of model parameters and facilitates the optimization process towards more generalizable local minima.

Furthermore, we have conducted a comprehensive ablation study to dissect the individual components of Helen and assess their specific contributions to the optimizer's overall performance. This holistic exploration provides valuable insights into the inner workings of Helen and its capacity to enhance generalization in CTR prediction models.

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

**Table 5: Key Statistics of Three Benchmark Datasets. Datasets are preprocessed by BARS, which includes the filtering of infrequent categorical features.**

| Dataset | #Instances | #Fields | #Features | %Positives |
|---------|-----------|---------|-----------|------------|
| Avazu | 40.43M | 24 | 3.75M | 16.98% |
| Criteo | 45.84M | 39 | 0.91M | 25.62% |
| Taobao | 25.03M | 20 | 2.62M | 5.15% |

## A APPENDIX

### A.1 Pseudocode of Helen

For a practical and comprehensive understanding, we present the pseudocode of the Helen optimizer here.

---

**Algorithm 1** Helen

---

**Require:** number of steps $T$, batch size $b$, learning rate $\eta$, perturbation radius $\rho > 0$, lower-bound $\xi$

1: **for** $t \leftarrow 1$ to $T$ **do**
2:     Draw $b$ samples $\mathcal{B}$ from $\mathcal{S}$
3:     $g \leftarrow \frac{1}{b} \sum_{x \in B} \nabla_w \mathcal{L}_\mathcal{B}(w)$
4:     $\hat{\epsilon}(w) \leftarrow [\ \rho \cdot \frac{g_h}{\|g_h\|}\ ]$       // Dense weights perturbation
5:     **for** each field $j$ and each feature $k$ in the field **do**
6:       $N_k^j(\mathcal{S}) \leftarrow \sum_{i=1}^n x_i^j[k]$
7:       $\rho_k^j \leftarrow \rho \cdot \max\{\frac{g_k^j}{\|g_k^j\|}, \xi\}$
8:       $\hat{\epsilon}(e_k^j) = \rho_k^j \cdot \max\{\frac{g_k^j}{\|g_k^j\|}, \xi\}$    // Embedding perturbation
9:       Append $\hat{\epsilon}(e_k^j)$ to $\hat{\epsilon}(w)$
10:     **end for**
11:     Compute Helen gradient $g^{Helen} = \nabla_w \mathcal{L}_\mathcal{B}(w)|_{w+\hat{\epsilon}(w)}$
12:     $w \leftarrow w - \eta \cdot g^{Helen}$
13: **end for**

---

### A.2 Dataset Description

We present essential statistics for three datasets in Table 5.

### A.3 Overall Performance On The Taobao Dataset

Table 6 summarizes the overall performance of seven models trained with different optimizers on the Taobao dataset.

### A.4 Hessian Eigenspectrum Analysis

Figure 6 visualizes the relationship between the top eigenvalue of the Hessian matrix and the frequency of the features on all three datasets.

### A.5 CTR Prediction Models Description

- **DNN**: DNN [13] pioneers a large-scale recommendation system powered by deep neural networks. In contrast to traditional linear models, DNN employs a fully-connected network to create dense vector representations for users. Candidate items are selected via top-N nearest neighbor search, and a neural network, incorporating candidate representations and contextual data, produces the final predictions.
- **Wide&Deep**: Wide&Deep [12] innovatively combines the advantages of traditional shallow linear models and deep neural networks within a unified learning framework. It comprises two integral components: the Wide Component, featuring a generalized linear model with cross-product transformation, and the Deep Component, which is a feed-forward neural network.
- **PNN**: PNN [54] consists of three key elements: an embedding layer for learning distributed representations of categorical features, a product layer for capturing feature interactions, and final fully-connected layers to make the prediction. This streamlined architecture enhances its capacity for handling multi-field categorical data with minimal computational overhead.
- **DeepFM**: DeepFM [21] is a deep learning-based CTR prediction model that combines the power of factorization machines (FM) and deep neural networks (DNN). The FM component of DeepFM is responsible for learning the low-order feature interactions, while the DNN component is responsible for learning the high-order feature interactions.
- **DCN**: The DCN [66] introduces an additional cross-network that explicitly generates feature interactions alongside the standard DNN architecture. This approach applies feature crossing at each layer, enabling high-degree interactions across features without the need for manual feature engineering, all while introducing minimal extra complexity to the DNN model.
- **DLRM**: DLRM [48] utilizes an MLP for continuous feature processing, computes second-order interactions between categorical embeddings and dense features, and produces predictions through a sigmoid-activated MLP. It efficiently reduces dimensionality by only considering cross-terms created via dot-products between embedding pairs in the final MLP layer, resembling factorization machines.
- **DCNv2**: DCNv2 [67] builds upon the straightforward cross-network architecture of DCN. It enhances this architecture by integrating it with a deep neural network to facilitate the discovery of complementary implicit interactions. Moreover, DCNv2 incorporates low-rank techniques and adopts the Mixture-of-Expert architecture [28, 62] to enhance computational efficiency and reduce cost.

### A.6 Optimizers Description

We will cover the mentioned seven optimizers in detail in this section.

- **Adam**: Adam [33] is a highly acclaimed optimization algorithm in the field of deep learning, renowned for its robustness and effectiveness. This algorithm combines the advantages of RMSprop [25] and Momentum [52] by tracking the moving averages of both first-order and second-order gradient moments to dynamically adapt the learning rate, making it a pivotal tool in the training of deep neural networks.
- **Nadam**: Nadam [16] is an extension of Adam optimizer, further enhances the Adam optimizer by replacing momentum component with Nesterov's accelerated gradient (NAG) [49], which has

**Table 6: Overall performance on the Taobao dataset.**

| Model | Adam LogLoss | Adam AUC | Nadam LogLoss | Nadam AUC | Radam LogLoss | Radam AUC | SAM LogLoss | SAM AUC | ASAM LogLoss | ASAM AUC | Helen LogLoss | Helen AUC |
|---|---|---|---|---|---|---|---|---|---|---|---|---|
| DNN | 19.529 | 63.520 | 19.373 | 63.261 | 19.496 | 63.183 | 19.433 | 63.532 | 19.575 | 63.528 | **19.390** | 63.620 |
| WideDeep | 19.433 | 63.570 | 19.391 | 63.140 | 19.423 | 62.960 | 19.582 | 63.053 | 19.566 | **63.704** | 19.437 | 63.691 |
| PNN | 19.366 | 63.660 | 19.411 | 63.303 | 19.420 | 63.109 | 19.434 | 63.036 | **19.357** | **63.754** | 19.368 | 63.711 |
| DeepFM | 19.481 | 63.486 | 19.449 | 63.249 | 19.457 | 63.079 | 19.436 | 63.473 | 19.461 | 63.385 | **19.441** | 63.752 |
| DCN | 19.507 | 63.052 | 19.403 | 63.148 | 19.481 | 63.016 | 19.502 | 63.214 | 19.498 | 63.111 | **19.354** | 63.753 |
| DLRM | 19.619 | 63.209 | 19.443 | 63.398 | 19.440 | 63.160 | 19.493 | 63.388 | 19.652 | 63.244 | **19.376** | 63.802 |
| DCNv2 | 19.489 | 63.059 | 19.383 | 63.455 | 19.490 | 63.191 | 19.452 | 63.199 | 19.474 | 63.135 | **19.343** | **63.848** |
| *avg* | 19.489 | 63.365 | 19.408 | 63.279 | 19.458 | 63.100 | 19.476 | 63.271 | 19.512 | 63.409 | **19.387** | **63.740** |

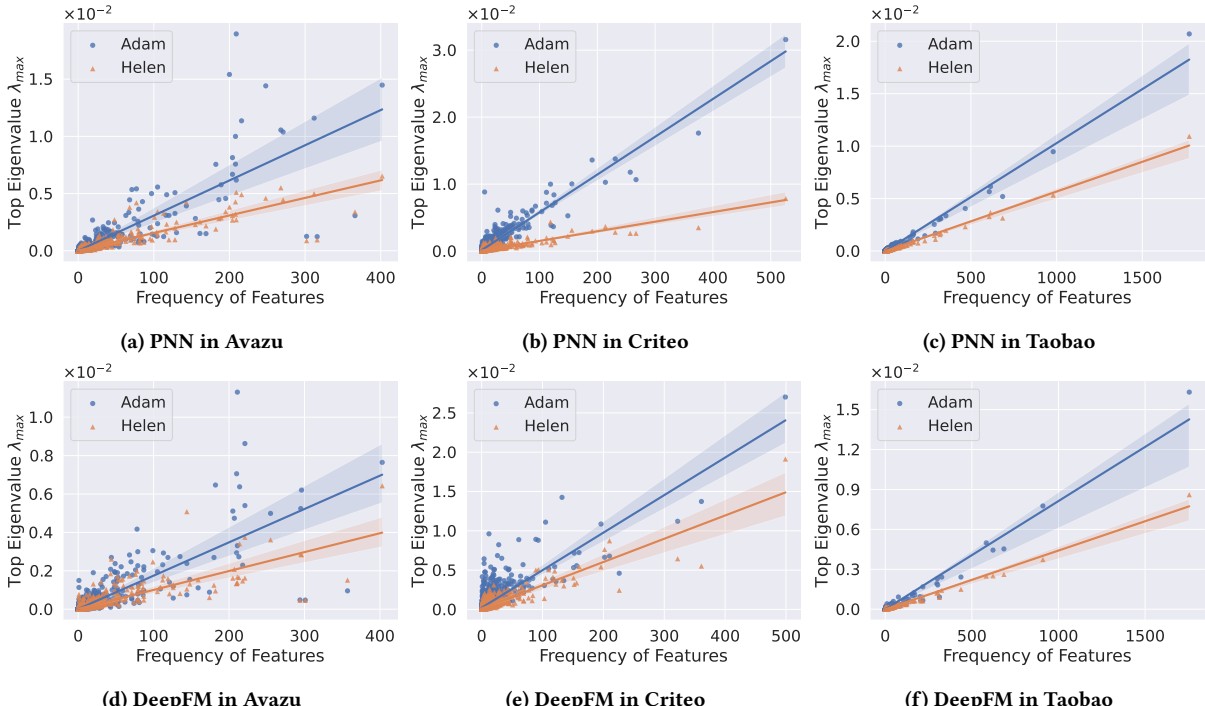

**Figure 6: Dominate Hessian eigenvalues of PNN and DeepFM in three datasets trained with Adam and Helen.**

a provably better bound. This adaptation allows Nadam to make more informed updates to model parameters and accelerates convergence, making it a robust and efficient choice for training neural networks.

- **Radam**: Radam [43], or Rectified Adam, is a variant of Adam optimizer, by introducing a term to rectify the variance of the adaptive learning rate, especially in the early stage of the training process. Its motivation comes from the consistent improvement of the warmup training strategy, which is widely used in the training of deep neural networks.

- **SAM**: SAM [20] represents a recent advance in enhancing the generalization performance of neural networks. SAM takes into account the geometry of the loss landscape by minimizing the loss function with respect to perturbed model parameters. This approach is equivalent to optimizing the loss function while

concurrently penalizing a measure of the model's sharpness without calculating the Hessian matrix, which is computationally expensive.

- **ASAM**: ASAM [34] stands as an adaptive iteration of SAM, tailored to dynamically fine-tune the perturbation radius for each layer's parameters. ASAM is claimed to possess scale-invariant property, achieved by adjusting the perturbation radius in relation to the weights' scale. This adaptive adjustment consistently yields improved generalization performance across various neural network architectures and tasks.

