# OpenReview forum: "Helen: Optimizing CTR Prediction Models with Frequency-wise Hessian Eigenvalue Regularization"
_ACM.org/TheWebConf/2024/Conference — TheWebConf24_

### Official Review · Reviewer_EjX3 · 2023-11-13

**Novelty:** 5
**Technical Quality:** 4

**Review:**

Summary:
In this paper, authors pointed out that CTR Models often have skewed feature distribution, and previous works reveal 1) the largest eigenvalue 𝜆 of the Hessian matrix 𝑯 of the loss function LS (·) indicates whether the optimization is ill-conditioned (line 316 to 321); 2) SAM has ability to effectively reduce the dominant eigenvalue of the Hessian matrix (line 396 to 401). The authors further validated that features of different frequencies have varying magnitudes of the largest eigenvalue of the Hessian matrix of the loss function, and there exists a strong linear correlation between the largest eigenvalue and feature frequencies. The authors also pointed out that SAM's uniform treatment of all weights may result in the largest eigenvalues of the Hessian matrix for some features being either too large or too small, which is detrimental to the optimization process. Therefore, the authors designed a variant of the SAM method, named Helen, to achieve frequency-wise hessian eigenvalue regularization. The authors conducted experiments on several CTR benchmarks to verify the Helen method.

Strength：
1. CTR prediction is an important problem for online recommendation and advertising, thus the paper's content is accessible to a wide audience.
2. The authors bridge the feature frequency-aware optimization and the research of sharpness-Aware minimization, and provide a new sharpness-aware frequency-wise minimization method for CTR Prediction tasks. It should also be suitable for other tasks that have skewed feature distribution.

Weakness：
1. Helen is designed for the tasks which have skewed feature distribution, if the feature distribution is close to uniform, Helen will degrade into SAM. So it's important to show the relationship between the improvement of Helen and the feature distribution of the benchmarks, but the paper has not presented this.
2. The description of some key points is unclear or wrong. See "Question 3" and "Typos or misstatements 1".
3. The baselines are not sufficient for this work. The paper mentioned that Cowclip first considers the impact of skewed feature distribution from the perspective of optimization. But Cowclip is not compared in the experiments, without reasonable explanation. I think it is very important to compare Cowclip to Helen. Another relatively minor flaw is that the paper does not compare with the latest WSAM method (Yue Y, Jiang J, Ye Z, et al. Sharpness-Aware Minimization Revisited: Weighted Sharpness as a Regularization Term. KDD2023: 3185-3194.), which is also a variant of SAM.

Typos or misstatements:
1. In Table 2, Helen is no better than Adam on AUC when the model is PNN, but the number of Helen is in bold. A similar mistake also occurred in Table 3, the AUC scores on PNN. By the way, it would be better to describe more about the information of the table in the caption, such as: what do the bolded terms in the table signify?

Other suggestions:
1. Table 5 is not appropriately positioned in the appendix. Placing it below Appendix A.2 would be more convenient for readers to access and read.

Missed References:
1. Yue Y, Jiang J, Ye Z, et al. Sharpness-Aware Minimization Revisited: Weighted Sharpness as a Regularization Term. KDD2023: 3185-3194.

**Questions:**

1. In section 2.1 (line 162), the features of the ctr model are formalized to contain only sparse features? Is it the proposed method applicable to models that include dense features?
2. I am a little confused about the relationship between the field and feature (line 162 to 174, line 325 to 348). The field means the key of an attribute such as age, and its feature means the value of age such as 20? If my understanding is wrong, can you give an concrete example in CTR task?
3. $\xi$ and $\rho$ seems to be hyper-parameters of Helen. Then how the feature frequency (N) influence the Helen method? I have not found the answer in Section 3.2 and Algorithm 1.

**Reviewer Confidence:**

3: The reviewer is confident but not certain that the evaluation is correct

**Scope:**

4: The work is relevant to the Web and to the track, and is of broad interest to the community

---

### Official Review · Reviewer_R4wG · 2023-11-23

**Novelty:** 5
**Technical Quality:** 3

**Review:**

Summary:

This paper reveals a consistent pattern in CRT prediction models, establishing a robust positive correlation between feature frequencies and the top Hessian eigenvalues linked to each feature. Leveraging this discovery, the paper introduces a novel optimizer, Helen, designed to prioritize the regularization of top Hessian eigenvalues of embedding parameters based on their respective feature frequencies.

Pros:
1. The discovery of the consistent pattern in CRT predictive modelling is insightful.
2. The paper is generally well-written and easy to follow.

Cons:
1. The experimental effect is not significant, with only a minor improvement observed.
2. Formatting errors are present, particularly in Table 4, where sign inconsistency errors are noted.

**Questions:**

How does the difference in the lower-bound parameter impact the final result?

**Ethics Review Description:**

No ethical issue

**Reviewer Confidence:**

3: The reviewer is confident but not certain that the evaluation is correct

**Scope:**

3: The work is somewhat relevant to the Web and to the track, and is of narrow interest to a sub-community

---

### Official Review · Reviewer_CDFK · 2023-11-25

**Novelty:** 3
**Technical Quality:** 4

**Review:**

In this paper, the authors approach the problem of click-through rate (CTR) prediction from an optimization perspective. By examining the characteristics of the data and optimization statistics related to CTR prediction, they show that there is a remarkable positive correlation between the top Hessian eigenvalue and feature frequency. This correlation suggests that frequently occurring features tend to converge towards sharp local minima, resulting in sub-optimal performance. To solve this problem, they adjust the perturbation radius using frequency information.
They used 7 different models in their experiments and compared the results with other optimizers such as SGD, ADAM, ... etc. on three different datasets.
The problem is clearly explained with the necessary statistical figures and citations. I must also say that the paper is clear and well-written.
I do not consider the novelty presented in the paper to be sufficient.
Despite the limited increase in AUC in the CTR domain and the apparent inadequacy of differences among other optimizers, I do not find the results of the proposed Helen optimizer to be significant. The observed improvement does not necessarily indicate the resolution of the local minima problem mentioned in the paper. Additionally, I believe it is crucial to investigate the impact of the proposed method and the compared optimizers on the training processes. It is necessary to demonstrate whether the loss function of the proposed method converges faster or slower than others.

**Questions:**

Did you conduct experiments during the training period to compare the proposed method with existing optimizers in terms of convergence?

How many times did you perform your experiments?

**Ethics Review Description:**

No.

**Reviewer Confidence:**

3: The reviewer is confident but not certain that the evaluation is correct

**Scope:**

4: The work is relevant to the Web and to the track, and is of broad interest to the community

---

### Official Review · Reviewer_AfeV · 2023-11-29

**Novelty:** 6
**Technical Quality:** 4

**Review:**

The paper addresses the problem of click-through rate (CTR) prediction in online advertising and recommendation scenarios. The authors propose a novel optimizer named Helen, which incorporates frequency-wise Hessian eigenvalue regularization to improve the performance of CTR prediction models. The paper highlights that feature frequency is strongly positively correlated to the top Hessian eigenvalue. This correlation suggests a higher likelihood of converging towards sharp local minima for more frequently occurring features, potentially resulting in sub-optimal outcomes. To overcome the issue, the authors propose to regularize the Hessian eigenvalues based on the feature frequency and achieve this by modifying the existing SAM optimizer. The paper maintains a clear and easily understandable flow. The empirical evidence effectively supports and justifies the key design choices. The proposed method Helen, building upon SAM's rationale, appears logically robust. The experiments substantiate the paper's claims to some extent, demonstrating its superiority. Despite these strengths, the paper exhibits weaknesses in certain experiments, writing, and technical aspects. These limitations affect the final assessment, as elaborated below:

W1. Equation (7) is highlighted as the pivotal component in Helen's design, yet it appears to deviate from expectations. Despite anticipating its relevance to frequency based on the context, the equation does not align with this description. As a result, the equation's intended purpose remains elusive, impeding a comaprehensive grasp of the detailed methodology.

W2. Given the minimal performance differences among various optimizers, conducting significance tests for each backbone model across individual datasets is important. Furthermore, there is a misrepresentation in the highlighting of the best performance of PNN, as the best one should be Adam instead of Helen about AUC.


W3.In Section 3.3, the primary drive behind Hessian eigenvalue regularization is elucidated as a means to augment generalization. Given the correlation between more frequent features and higher Hessian eigenvalues, does this imply that the performance of CTR prediction is constrained by the samples with more frequent features? It would be beneficial to conduct performance analyses on groups categorized by varying feature frequencies to explore this aspect further. That is, providing evidence for that frequently occurring features tend to converge towards sharp local minima.

W4. Regarding the baseline, given that Cowclip also accounts for the skewed feature distribution during optimization, why hasn't it been included for comparison?

W5.The experiment conducted involved tuning hyperparameters within a narrow range. Could this limitation potentially impact the conclusions drawn in this paper? It's plausible that distinct optimizers might necessitate significantly different hyperparameters for optimal performance.

W6. How does the training cost of Helen compare to other methods?

****************************************************************
I have read the rebuttal.

**Questions:**

See the questions listed in the Review section.

**Ethics Review Description:**

-

**Reviewer Confidence:**

3: The reviewer is confident but not certain that the evaluation is correct

**Scope:**

4: The work is relevant to the Web and to the track, and is of broad interest to the community

---

### Decision · Program_Chairs · 2024-01-22

**Decision:**

Accept

**Comment:**

Summary: this paper introduces Helen, a novel optimizer tailored for Click-Through Rate (CTR) prediction models. It focuses on addressing the skewed feature distribution problem commonly encountered in CTR models by optimizing frequency-wise Hessian eigenvalue regularization. This paper presents empirical validations across multiple CTR benchmarks, demonstrating Helen's effectiveness in reducing the largest eigenvalues of the Hessian matrix and enhancing model performance.

 #### Strengths
 1. **Innovative Approach**: Helen's integration of feature frequency into SAM optimization for CTR prediction models is a novel and logical advancement.
 2. **Empirical Validation**: The effectiveness of Helen is validated across several benchmarks, showcasing its ability to improve CTR prediction performance.
 3. **Relevance**: The paper addresses an important problem in CTR prediction, namely skewed feature distribution, which has a significant impact on model optimization.

 #### Weaknesses
 1. **Minor Performance Improvements**: The AUC improvements shown by Helen are relatively small, raising questions about its practical significance.
 2. **Clarity and Technical Details**: The paper has some unclear descriptions, particularly in the formulation of Helen and its relationship with feature frequency.
 3. **Lack of Comprehensive Baseline Comparison**: The absence of comparisons with other skew-aware optimization methods like Cowclip is a missed opportunity for a thorough evaluation.

 #### Suggestions for Improvement
 1. **Clarify Technical Aspects**: Provide a clearer explanation of how Helen's formulation directly relates to feature frequency and its impact on optimization.
 2. **Enhance Performance Evaluation**: Conduct significance tests to validate the practical significance of the observed AUC improvements.
 3. **Include more Baseline Comparisons**: Include comparisons with other skew-aware optimization methods like Cowclip and the latest variants of SAM for a more comprehensive evaluation.